# SINE-Based Phylogenomics Reveal Extensive Introgression and Incomplete Lineage Sorting in *Myotis*

**DOI:** 10.3390/genes13030399

**Published:** 2022-02-23

**Authors:** Jennifer M. Korstian, Nicole S. Paulat, Roy N. Platt, Richard D. Stevens, David A. Ray

**Affiliations:** 1Department of Biological Sciences, Texas Tech University, Lubbock, TX 79409, USA; j.korstian@ttu.edu (J.M.K.); nicole.paulat@ttu.edu (N.S.P.); 2Host-Pathogen Interactions Program, Texas Biomedical Research Institute, San Antonio, TX 78227, USA; neal.platt@gmail.com; 3Department of Natural Resource Management and Natural Science Research Laboratory of the Museum of Texas Tech, Texas Tech University, Lubbock, TX 79409, USA; richard.stevens@ttu.edu

**Keywords:** retrotransposons, phylogenetics, introgression, incomplete lineage sorting

## Abstract

Using presence/absence data from over 10,000 Ves SINE insertions, we reconstructed a phylogeny for 11 *Myotis* species. With nearly one-third of individual Ves gene trees discordant with the overall species tree, phylogenetic conflict appears to be rampant in this genus. From the observed conflict, we infer that ILS is likely a major contributor to the discordance. Much of the discordance can be attributed to the hypothesized split between the Old World and New World *Myotis* clades and with the first radiation of *Myotis* within the New World. Quartet asymmetry tests reveal signs of introgression between Old and New World taxa that may have persisted until approximately 8 MYA. Our introgression tests also revealed evidence of both historic and more recent, perhaps even contemporary, gene flow among *Myotis* species of the New World. Our findings suggest that hybridization likely played an important role in the evolutionary history of *Myotis* and may still be happening in areas of sympatry. Despite limitations arising from extreme discordance, our SINE-based phylogeny better resolved deeper relationships (particularly the positioning of *M. brandtii*) and was able to identify potential introgression pathways among the *Myotis* species sampled.

## 1. Introduction

Two key factors have been identified as drivers of discordance when inferring phylogenies of rapidly diverged taxa: Incomplete Lineage Sorting (ILS) and gene flow between species (hybridization or introgression). ILS occurs when relatively young, polymorphic traits are passed down inconsistently from the parent species to the newly arisen species. ILS is expected to be more prevalent in species with histories of rapid adaptive radiations since there is less time for polymorphic insertions to become fixed. This inconsistent sorting often confounds efforts to reconstruct an accurate species tree. Likewise, the exchange of genetic material among species by introgression or hybridization can create contrary evolutionary signals.

Moreover, the relationships recovered by analyses relying on nucleotide sequences can recover radically different topologies if the evolutionary history of the selected genes or regions are sufficiently distinct. Different coding regions, for example, may experience different selective pressures which could then cause those regions to produce phylogenetic patterns that disagree with each other or even with the overall species tree. Efforts have been made to overcome this shortcoming by analyzing multiple genes and analyzing the concatenated genic sequences, thus increasing the number of loci and genomic regions used to infer the phylogenies.

Because they do not appear to share the same susceptibility to these problems as sequence-based methods, transposable elements (TEs) have gained popularity as markers for phylogenomic studies [1,2,3,4,5]. TEs are selfish genetic elements that can move and/or replicate within a genome. The “copy and paste” mechanism of transposition utilized by retrotransposons, such as Short INterspersed Elements (SINEs), often results in a pattern of abundant insertions interspersed throughout the genome. The presence/absence patterns of SINE insertions have been used to untangle some of the more contentious relationships among basal mammal clades [1,2,3,4] and has been suggested as a “nearly perfect character” for phylogenetic reconstruction [6]. SINEs are abundant in many mammal genomes, and typically experience very low levels of homoplasy because they rarely experience precise excisions or parallel insertions [1,2,6,7].

Because the insertion mechanisms of SINEs are well characterized, it is possible to assume that the absence of a given insertion is the ancestral state for all species, therefore the presence of an orthologous insertion in two species indicates that the insertion originated in a common ancestor for those two species. This phenomenon means that in addition to their use for phylogenetic reconstruction, SINEs and other retrotransposons can be used to investigate the frequency and duration of ILS events. In birds, for example, an examination of presence/absence patterns of over 3000 Long Terminal Repeat (LTR) insertions corresponded to the neoavian radiation and made it possible to calculate the rate and duration of ILS experienced during that radiation [8]. Large scale retrophylogenomic studies have provided compelling evidence that multiple species arising simultaneously during a single speciation event may be more widespread than previously thought [4,8]. 

Several studies of Chiroptera using sequence information from relatively few loci have helped inform bat phylogenies across a broad taxonomic range [9,10,11,12,13,14]. A recent study of the genus *Myotis* sequenced full mitogenomes and thousands of UltraConserved Elements (UCEs; regions of high conservation across a wide range of mammalian taxa) and found that the evolutionary histories recovered with the two different data sources differed significantly (Figure 1) [15].

The genus *Myotis* is currently represented by 131 species, can be found on every continent except Antarctica, occupies a variety of ecological niches, and exhibits several dietary specializations [16]. Molecular phylogenetics divides *Myotis* into two major clades and a number of subclades: a New World clade (with three subclades) and an Old World clade (with an Ethiopian clade and ≥8 Eurasian clades) [12]. Subsequent radiations within the New World clade are split into the Neotropical and Nearctic subclades plus an enigmatic clade that includes two Old World species [12]. The unusual placement of this enigmatic clade (which includes *M. brandtii*) suggests that lineage sorting and/or introgression is occurring within this genus.

Indeed, hybridization between extant species of *Myotis* is known to occur and may be extensive within the genus, though its current extent is unknown. For example, after decades of debate, a recent study proposed that *M. lucifugus* and its four subspecies should actually be categorized as five paraphyletic, non-sister species with ongoing gene flow among species in areas where their ranges overlap [17]. Hybrids of *M. blythii* and *M. myotis* have been identified using fecal DNA in a maternity colony shared by both species [18]. Hybridization was proposed as a potential explanation for individuals with intermediate phenotypes in areas where *M. lucifugus carissima* and *M. yumanensis sociabilis* are sympatric [19]. *M. myotis* and *M. oxygnatus* hybridization has been confirmed using both morphological and genetic evidence [20]. The high degree of morphological similarity within the genus has made identification of hybrids on the basis of morphology alone extremely challenging, which has kept the extent and impact of hybridization within the genus relatively uncharacterized. Genetic evidence, such as phylogenomic reconstructions using retrotransposons, offer a valuable opportunity to investigate the impacts of introgression in *Myotis*.

This study utilizes low-coverage sequence data to identify SINE insertion patterns across 11 *Myotis* species. We used two methods to infer a SINE-based phylogeny, which was then used to clarify evolutionary relationships among species, identify loci discordant with the overall species tree, and evaluate the relative contributions of ILS and hybridization in these species. For comparison with the retrotransposon-based phylogeny, we also generated a tree based on UCEs. Our results indicate that both ILS and gene flow among species are abundant within *Myotis*.

## 2. Materials and Methods

We selected single representatives from eight New World *Myotis* species for low-coverage whole genome sequencing (WGS): *M. septentrionalis*, *M. austroriparius*, *M. occultus*, *M. ciliolabrum*, *M. vivesi*, *M. velifer*, *M. thysanodes*, and *M. yumanensis* (Table 1). Previously published WGS read data for two representatives of Old World species, *M. brandtii* and *M. davidii,* were obtained from GenBank [21,22]. These species were selected based on availability and to encompass a wide range of taxonomic relatedness. *M. lucifugus* and *M. occultus*, for example, are so closely related that there are disagreements about whether they are non-sister species, subspecies, or sister species [12,15,17,23,24,25]. Of the two species from the Old World (*M. davidii* and *M. brandtii*), genetic studies consistently place *M. davidii* outside the New World clade, while the phylogenetic placement of *M. brandtii* has been inconsistent, appearing basal to the New World clade in some studies and within the clade in others [9,11,12,15].

Tissue samples for this project came from the Texas Tech Natural Science Research Laboratory (Table 1) and were collected under the auspices of the TTU IACUC protocol 15010-01. Genomic DNA was extracted from tissue samples using the Qiagen DNEasy extraction kit protocol. Sequence libraries were prepared using the Nextera DNA library prep kit from Illumina (San Diego, CA, USA) and sequenced as paired-end 150 bp reads on a HiSeq 2500 at the Center for Biocomputing and Genomics at Texas Tech University. All WGS reads were quality filtered and trimmed using Trimmomatic v0.27 [26] and mapped to the *M. lucifugus* genome (Myoluc2.0, GCA_000147115.1; [27] with BWA-MEM v0.7.17 [28]. The resulting BAM files were merged with Picard v2.5.0 [29] and sorted with Samtools v1.9 [30] before read groups were added and duplicates removed using Picard.

The Mobile Element Locator Tool (MELT) v. 2.0.2 [31] locates and annotates not only known TE insertions found from the reference genome, but also non-reference insertions. MELT locates TE insertions by finding split and discordant read pairs and creating a list of putative non-reference TE insertions based on the number of reads supporting an insertion and to genotype non-reference species for known insertions in the reference genome. An existing species-specific TE library for *M. lucifugus* [32,33] was used to identify insertions of the most abundant SINE family in *Myotis*, Ves, in each genome using RepeatMasker [34]. RepeatMasker output was converted into BED format using the RM2Bed.py utility [35]. MELT analysis was conducted in both Split and Deletion modes for all samples. We included the consensus sequences from each of the Ves subfamilies in the *M. lucifugus* TE library for our search and allowed for 3% mismatches in identified insertions.

To assess the accuracy of the MELT insertion calls, we also created synthetic paired-end 150 bp Illumina reads from the published *M. lucifugus* genome using SimSeq [36] at approximately 15x coverage. The synthetic reads were subjected to the same MELT analysis as the other samples. 

MELT output VCF files were filtered for quality to include all orthologous TE insertion calls from Split mode receiving “PASS” scores in “filter” column and “ASSESS” values >1 in the “info” column. We then combined the reference (Split mode) and non-reference (Deletion mode) MELT output and converted into presence/absence genotypes for all Ves insertions for all 11 *Myotis* species and the *M. lucifugus* synthetic reads. A heterozygous genotype was considered a ‘presence’ for that insertion in that species. Heterozygosity was calculated from the number of heterozygous calls in the VCF output.

For phylogenomic analyses of the TE dataset, we only examined Ves insertions where genotypes were determined for all 11 species (i.e., no missing data was permitted) and a matrix of binary genotypes was constructed. Three methods were used to infer phylogenetic relationships from this matrix: (1) the Dollo parsimony method from [8], (2) using ASTRAL to infer gene trees from bipartition trees (referred to as ASTRAL_BP) from [37], and (3) a neighbor-joining tree and principal component analysis.

We selected Dollo parsimony because it is the non-coalescent evolutionary model that best matches TE insertion dynamics by only allowing character states to change from ”0” (absence) to “1” (presence). In the Dollo parsimony analysis, we filtered the Ves loci to retain only those that were potentially phylogenetically informative. We used the *seqboot* module in PHYLIP v3.695 [38] to generate 1000 bootstrap replicates of our dataset. Then, we ran the *dollop* module in PHYLIP on the bootstrap datasets with the following options (Dollo parsimony, jumbling sequence order 7 times, a random number seed of 1111, and state transitions generated) to generate a tree for each bootstrapped dataset. We then determined the majority-rule consensus tree from the bootstrap trees using *consense* modules of PHYLIP. The tree was visualized using the Interactive Tree Of Life iTOL [39]. A second analysis with the *dollop* module to compare the 01 insertion matrix to the consensus tree provided us with the number of state transitions required for a given locus to be consistent with the consensus tree. Following the methods from [8], we classified loci with 2+ state transitions in the *dollop* output as discordant loci. Discordant loci were then mapped onto the consensus tree to determine when the ILS events likely occurred. Each discordant locus was attributed to a potential ILS event that occurred in the shared common ancestral population of all species possessing that insertion.

The ASTRAL bipartion method (ASTRAL_BP) was created as an “ILS aware” methodology to infer phylogenies from retroelement insertion patterns in situations where ILS is prevalent (as it appears to be in *Myotis*) [37]. This method takes the same binary genotype matrix used in the Dollo method and creates an incompletely resolved gene tree with exactly one bipartition for each locus using a custom script (http://github.com/ekmolloy/phylotools; accessed on 21 August 2021) [37]. ASTRAL-MP v5.15.4 [40] was then used to infer a species tree which was visualized using iTOL. 

As a complement to the majority-rule consensus tree, a principal component analysis (PCA), we conducted and inferred a neighbor-joining (NJ) phylogeny from the 01 matrix with the *adegenet* package in R v3.6.1 [41]. We created phylogenetic median joining networks using SplitsTree v4.14.4 [42] and a set intersection plot of discordant loci using the *UpSetR* package in R [43].

To assess the extent of introgression within our samples, we used a quartet asymmetry test designed specifically for use with retrotransposon insertions [37]. We created a list of all 330 possible 4-species quartet combinations from our 11 species. Because previous analyses had been conducted with fewer quartet combinations, we created custom scripts to implement the methods described in [37]. We used a nexus file to perform a series of commands for each possible quartet using PAUP* v 4.0a169 [44] (an example of these nexus files is available on https://github.com/davidaray/retrophylogenomic_tools; accessed on 21 August 2021). In brief, this nexus command file reads the “01” insertion matrix, deletes all species not in the quartet of interest (the delete command), prunes the ASTRAL-BP tree to include only the quartet species, excludes all non-informative characters (using the exclude command), performs a heuristic parsimony search for the 3 best trees, scores those trees, and writes the output and trees to several files. Next, a custom python script (available on https://github.com/davidaray/retrophylogenomic_tools) creates a table by extracting the number of informative characters and tree lengths (also referred to as number of steps and tree scores) for each of the 3 trees, and identifying the “accepted” topology consistent with the ASTRAL-BP tree. With these files in hand, we then calculated the number of characters supporting each quartet configuration as described in [37], which is equal to the total number of informative characters minus the number of characters that do not support the quartet. Number of characters not in support is calculated by subtracting the total number of informative characters from the quartet tree length. We calculated the expected number of characters supporting the two alternate splits for each quartet by assuming all characters not supporting the “accepted” topology would be equally divided between the two remaining splits. A χ^2^ test was conducted to compare the observed and expected numbers of characters supporting the alternate splits. *p*-values were corrected for multiple comparisons using a Holm–Bonferroni sequential correction [45] with an Excel calculator [46]. For quartets with significant deviations from expectations under ILS, we determined which of the three splits was overrepresented and which two species were involved in the potential introgression event. 

Using de novo or reference-based assemblies for each species, we mined UCEs from each species, which were then used to infer an independent phylogenetic hypothesis using the PHYLUCE pipeline [47]. Within PHYLUCE, we retained the default values unless otherwise specified. In brief, the contigs of the assemblies were fragmented and matched to the UCE-5k-probe set provided with PHYLUCE. Genomic sequences matching probes plus 500 bp of flanking sequence were then extracted from the genomes, aligned using MAFFT [48], and edge trimmed. Only alignments containing sequences for all taxa were retained for further analysis. Individual gene-trees were inferred for each UCE locus with RAxML-NG [49] with a single analysis that implements a GTR+G mutation model and 100 bootstrap trees after selecting the tree with the maximum likelihood from the 10 parsimony-based starting tree topologies. In addition to a single best maximum likelihood (ML) tree, RAxML-NG also produces a tree where near-zero branches were collapsed. We used the collapsed versions of the best ML trees to infer the species tree using ASTRAL and the resulting tree was tested for hard polytomies [50,51]. This polytomy test, based on the multi-species coalescent model, examines whether a branch should be replaced with a polytomy. The resulting tree was visualized using iTOL [39].

## 3. Results

### 3.1. WGS Mapping and TE Variation Discovery

We mapped 10 WGS datasets from *Myotis* bats with a coverage depth between 14 and 30x to the *M. lucifugus* assembly (Table 1). From the mapped data, MELT called 13,392 non-reference Ves insertions (i.e., absent from the *M. lucifugus* reference genome) and obtained 202,145 genotypes for reference Ves insertions in the other 10 species. Of the 215,537 total Ves genotypes, 211,494 (98.1%) passed filtering. The total number of extracted Ves insertion calls per species after filtering ranged from 22,220 to 17,315. MELT called 100% of the reference insertions in the synthetic *M. lucifugus* reads and did not make any false positive calls on the non-reference insertions. Genotype calls for the synthetic reads are not included in the reported call totals throughout this manuscript. Heterozygous insertions were infrequent, and heterozygosity ranged from 0.0004 in *M. brandtii* to 0.0048 in *M. velifer*. 

### 3.2. Phylogenetic Relationships in Myotis

We identified 10,595 phylogenetically informative Ves insertions. By creating a presence–absence matrix from the insertion genotypes, we inferred phylogenetic trees using our four different methodologies (Figure 2A,B, Figure 3A and Figure 4A). Contrary to prior studies [11,12,14,15], all phylogenies inferred in this study consistently place *M. brandtii* outside the North American clade. This is further supported by the principal component analysis (Figure 3B) in which principal component one accounts for 41% of the variance in the samples and clearly discriminates between the Old World species, *M. davidii*, and the New World species with *M. brandtii* intermediate between the two extremes.

The Dollo parsimony tree (Figure 2A) does not provide meaningful branch lengths or any indication of the nature of the phylogenetic conflict at a given node and relies on bootstrap support instead. Although branches with more than 85% bootstrap support are shown in Figure 2A, given the amount of conflict revealed by the ASTRAL_BP trees, it seems a more reliable, realistic approach to SINE-based, Dollo parsimony trees would be to collapse all branches with less than 100% bootstrap support. If the branches with less than 85% support are collapsed in the Dollo tree, all trees return similar, though not identical, topologies.

Most trees produced in this study (Figure 2A,C, Figure 3A and Figure 4A) support dividing the North American species into three distinct clades each composed of three species. *M. thysanodes*, *M. septentrionalis*, and *M. ciliolabrum* form a clade. *M. occultus* and *M. lucifugus* are sister species with *M. vivesi* basal to that pair, rounding out the second clade. The third clade is composed of *M. austroriparius*, *M. yumanensis*, and *M. velifer*. In the principal component analysis (Figure 3B), principal component two, which accounts for 16% of the variance in the samples, effectively separates and supports the three distinct clades. Interestingly, the ASTRAL-BP tree (Figure 2B) yields conflicting measures of support for the three North American clades, with all branches receiving posterior probability scores of one despite having significant quartet support for the alternate topologies at many of the branches. In contrast, the ASTRAL-UCE tree appropriately assigns the lowest scores to the branches with the most phylogenetic conflict.

### 3.3. Ves Insertion Discordance

By comparing our matrix of Ves genotypes with the three main output trees, we located 3044 loci (28.7% of all informative loci) that were inconsistent with the Dollo tree (Figure 2A). In contrast, the ASTRAL-BP and ASTRAL-UCE trees both resulted in higher discordance counts (4830 and 4143 loci, respectively). We used the Dollo parsimony tree to determine the timing of the potential ILS/introgression events. We considered each discordant locus to be a signal for a potential ILS/introgression event in the shared common ancestor of all species possessing the insertion. For example, if the same insertion was present in *M. septentrionalis*, *M. yumanensis*, and *M. vivesi*, then the discordance most likely arose in the most recent common ancestor of those three species. This concept is illustrated in Figure 4C with the dot matrix plots showing discordant insertion patterns and their prevalence in our Ves dataset. The insertion patterns in Figure 4C are colored to correspond with their node of origin on the Dollo tree in Figure 4B. The majority (73.7%) of the discordant insertion patterns are localized to two main nodes (Figure 4B).

### 3.4. UCE Phylogeny

We extracted 2797 UCE loci for use in this study. The average length of the UCE sequences was 1147.8 bp (min: 655 bp, max: 1791 bp). In total, 3,173,968 bp of sequence was used to infer a UCE-based phylogeny based on the 13,873 phylogenetically informative sites within the sequences (Figure 2C). Four branches on the tree have posterior probabilities that are < 0.9, indicating poor support. A polytomy test for those four nodes (indicated with purple dots in Figure 2C) failed to reject the null hypothesis that the node is best represented by a polytomy. Collectively, the polytomy test and low support values suggest that although additional structure is shown in Figure 2C, the UCE tree is still consistent with the three main clades within the North American taxa but shows them arising from a single polytomy.

### 3.5. Evidence of Gene Flow

The quartet asymmetry test indicated that 183 of the 330 possible quartet combinations for our dataset showed significant preference for one of the two alternate topologies indicative of deviation from expectations under ILS alone (Table 2). A majority (57.2%) of the quartets exhibiting potential introgression involve *M. brandtii* or *M. davidii*, suggesting that extensive gene flow has occurred between the Old and New World *Myotis* populations. Based on quartet patterns, three separate, probable introgression events have been identified involving Old World species (Pathways I, II, IV, Figure 5). Five species pairs were heavily represented among the potentially introgressed quartet events suggesting recent or current hybridization: *M. ciliolabrum/M. thysanodes* (7 quartets), *M. lucifugus*/*M. septentrionalis* (9 quartets), and *M. austroriparius* paired with *M. velifer* (8 quartets), *M. lucifigus* (13 quartets), and *M. septentrionalis* (12 quartets). 

## 4. Discussion

Using presence/absence data from 10,595 Ves insertions, we reconstructed a phylogeny for 11 *Myotis* species. This phylogenetic framework will enable more refined interpretations of ILS and hybridization within *Myotis*. With 28.7% of individual Ves trees discordant with the overall species tree, phylogenetic conflict appears to be rampant in this genus. Phylogenetic conflict is extremely high in areas where branch lengths are short in the ASTRAL-UCE tree. This suggests that in *Myotis*, ILS may be a major contributor to the discordance because the short divergence times would not allow sufficient time for complete lineage sorting.

The observed phylogenetic conflict within *Myotis* could arise from ILS, hybridization, or phylogenetic error and it can be difficult to determine how much conflict is being contributed by each of those sources. For instance, the most common discordant insertion is present in all species except *M. brandtii* and *M. septentrionalis*, a pattern that occurred 163 times in our dataset. Since this pattern was recovered so frequently, it seems likely that this is a real pattern as opposed to genotyping or phylogenetic error. Previous studies have found that MELT accuracy with SINE insertions is high when provided with at least 5x sequencing coverage [4]. In one study, MELT was able to correctly locate SINE insertions 99% of the time in a simulated dataset and 92% of those insertions were correctly genotyped [4]. The synthetic *M. lucifugus* reads were scored by MELT without error in our experiment. These factors suggest that the false signals generated by MELT errors are unlikely to mask discordance patterns generated by ILS or introgression. 

Much of the discordance in the *Myotis* Ves tree can be attributed to two nodes, the oldest of which represents 39.6% of all potential ILS/introgression events and corresponds with the split between the Old World and New World *Myotis* clades that occurred approximately 18 MYA [12]. The younger node, which comprises 34.2% of the potential ILS/introgression events, corresponds with the first radiation of *Myotis* within the New World approximately 12 MYA [12]. Furthermore, the abundance of ILS in *Myotis* also suggests that historic populations were large, well-connected, and harbored high diversity.

Ruedi et al. [12] proposed that an ancestral *Myotis* species from the Eastern Palearctic expanded its range to the New World around 18 MYA. While all previous studies have placed *M. brandtii* within the New World clade, all of our trees place it with the Old World species, which is more consistent with its current range in Russia [22]. Our analysis revealed evidence of hybridization between the Old and New World species in this study, the extent of which suggests that secondary contact and hybridization between Old and New World species continued for some time after the initial vicariance event. Our study included just two representatives from the Old World, *M. brandtii* and *M. davidii*, both of which have current distributions within the Eastern Palearctic/Asia that could plausibly have extended ranges into or near North America during Miocene interglacial periods. Proposed introgression pathway I between *M. davidii* and the ancestor of all New World *Myotis* would be consistent with loss of contact and gene flow with *M. davidii*, potentially due to range contraction, approximately 12 MYA [12]. Proposed introgression pathways II and IV suggest that contact between *M. brandtii* and New World *Myotis* may still have occurred until approximately 8 MYA [12]. This continued gene flow between *M. brandtii* and the New Wold *Myotis* likely contributed to its ‘enigmatic’ placement in previous studies. Our introgression tests also revealed evidence of both historic and more recent, perhaps even contemporary, gene flow within *Myotis* species of the New World. Hybridization in mammals, once thought to be a rare occurrence, has been found by an increasing number of genomic studies [52]. Our findings suggest that hybridization not only happened in *Myotis*, but it likely played an important role in the evolutionary history of the genus and may still be happening in areas of sympatry as has been shown in European *Myotis* populations [18,20]. 

While their performance in cases with limited ILS and introgression has shown SINEs to be excellent characters for phylogenetic reconstructions, their performance is hampered when ILS and introgression both play a significant role in the organism’s evolutionary history as is the case with *Myotis*. Despite limitations arising from extreme discordance, our SINE-based phylogeny was able to better resolve deeper relationships (particularly the positioning of *M. brandtii*) and was able to identify potential introgression pathways among the *Myotis* species sampled. Retroelement insertion patterns have been successfully employed to decipher evolutionary relationships on a variety of scales from the genus level [4,5] up through the level of superorders in mammals [1,2,3,7,8]. Retrophylogenomics offers a data-rich option to determine genomic patterns of phylogenetic conflict that is less computationally intensive than whole genome alignments and analysis methods are now available that are statistically consistent even in anomaly zones [53]. TE datasets can be mined with relative ease from low-coverage sequencing projects that are already available or conducted as a complement to other ongoing projects without requiring a specific library preparation, high coverage sequencing, or de-novo genome assembly. 

## Figures and Tables

**Figure 1 genes-13-00399-f001:**
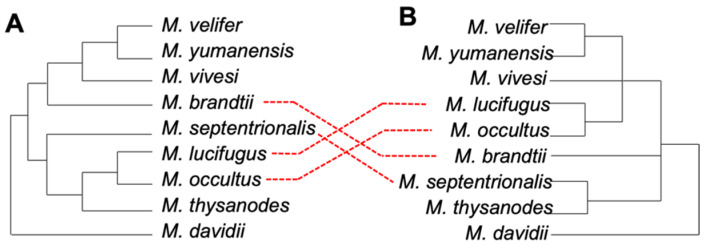
A simplified version of recent mitochondrial (**A**) and nuclear (**B**) phylogenetic trees for the genus *Myotis*. The trees were constructed using whole mitogenomes and 3648 nuclear UCEs. Adapted from [15] with permission.

**Figure 2 genes-13-00399-f002:**
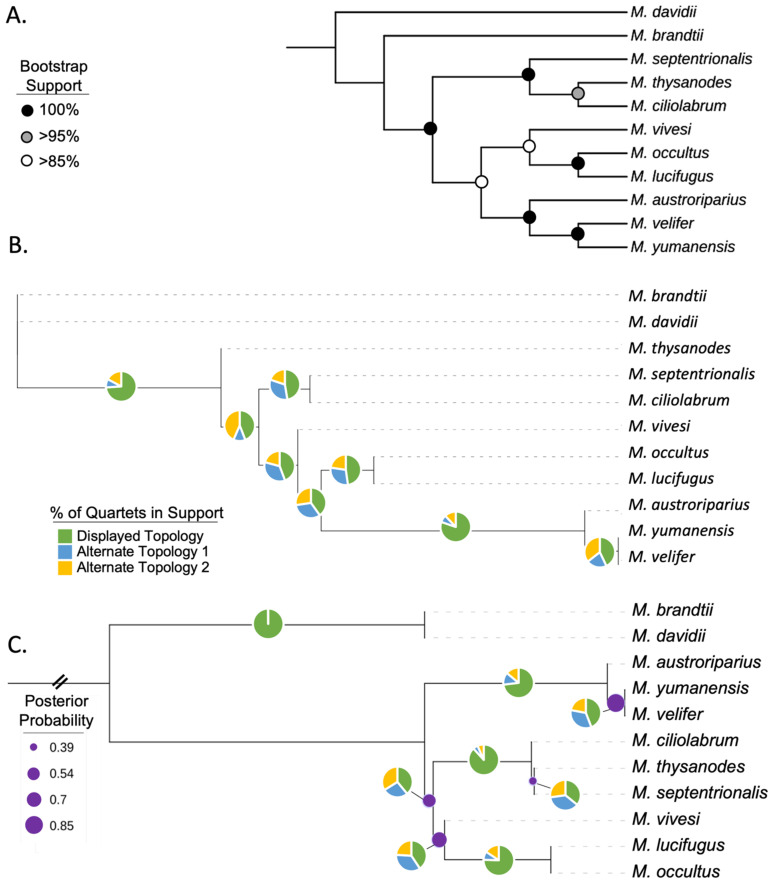
(**A**) Majority rule Dollo parsimony consensus tree constructed from 10,595 orthologous Ves insertions with 1000 bootstrap replicates. Circles on nodes represent bootstrap support values (black = 100%, grey > 95%, white > 85%). (**B**) Ves phylogeny inferred by ASTRAL using incompletely resolved bipartitions as gene-trees. All branches had posterior probability of 1. In Figure (**B**,**C**), pie charts on each branch show the % of quartets supporting the three topologies at that branch. (**C**) UCE phylogeny inferred by ASTRAL using gene trees of 2797 UCE loci. Purple dots on branches indicate *p*-values > 0.05 from polytomy testing (suggesting a polytomy would better represent this node) and are scaled by size according to posterior probability. The basal branch was truncated for better readability.

**Figure 3 genes-13-00399-f003:**
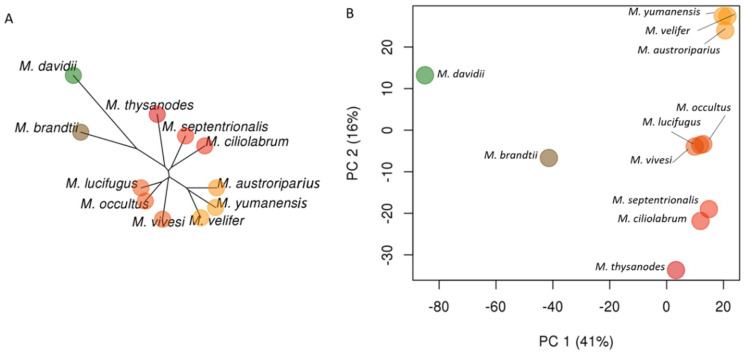
Genetic structure within *Myotis* as revealed by 10,595 orthologous Ves insertion events. (**A**) Unrooted neighbor-joining tree based on Ves insertions. (**B**) Scatterplot of the first two principal components (PC1 and PC2) illustrating differences among *Myotis* genomes. Percentages indicate percent of variation explained by each principal component shown. Colors in both figures are determined by PC values.

**Figure 4 genes-13-00399-f004:**
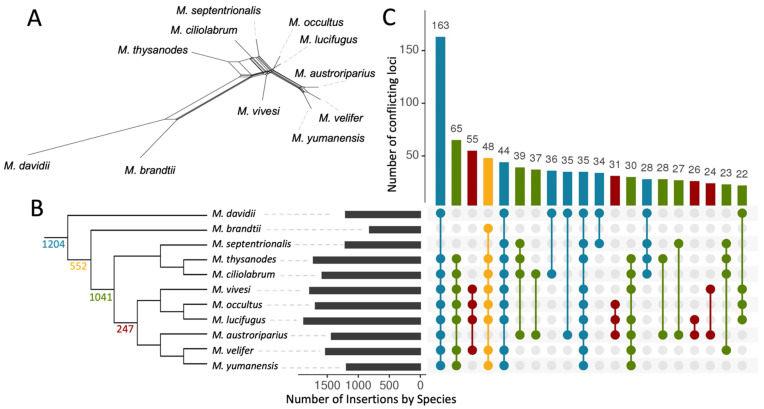
Visualizations of phylogenetic discordance among *Myotis* inferred from polymorphic Ves insertions. (**A**) Unrooted median joining network tree generated from all 10,595 loci. (**B**) Majority rule consensus tree illustrating the distribution of discordant signals in the dataset. Discordant Ves insertions (2+ state changes) are considered a phylogenetic signal for the common ancestor for all taxa carrying the insertions. Counts on the tree indicate the number of synapomorphies coalescing at that node. For nodes without counts, all insertions within that clade alone had fewer than 2 state changes and did not qualify as discordant. The colors and order of species is the same in B and C. (**C**) Characterization of 3044 discordant insertions. The black bars on the left show the number of discordant insertions present in each species. The dot matrix indicates the synapomorphic patterns, with each pattern represented by a vertical line with dots indicating presence of the insertion in that species. Colors in the dot matrix correspond to the node of coalescence for that insertion in B. The y-axis above gives a count of times that pattern occurred among all 3044 discordant loci. Only the top 20 most prevalent patterns are shown on this graph. See text for additional information.

**Figure 5 genes-13-00399-f005:**
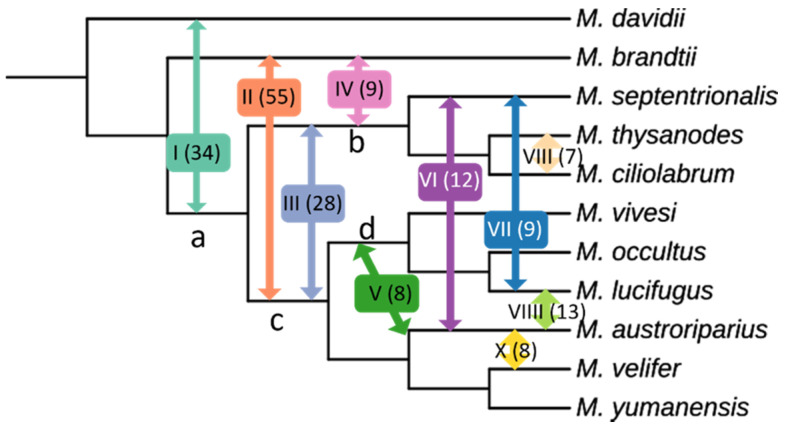
Proposed Introgression Pathways identified using quartet asymmetry tests. A total of 83 quartets indicated significant deviation from expectations under ILS. Pathways are identified by roman numerals with number of quartets supporting in parentheses.

**Table 1 genes-13-00399-t001:** Summary of *Myotis* whole genome sequences used in this project. The Cov. column refers to the average coverage of the *M. lucifugus* reference genome for that species. The Calls column refers to the number of Ves calls for each species after quality filtering. The accession column includes NCBI accession numbers.

Species	ID Number	Cov.	Accession	Calls	Heterozygosity
*M. austroriparius*	M8132	19.2x	SRR17555657	21,916	0.0038
*M. brandtii*	wgs *	16x	ANKR00000000	17,315	0.0004
*M. ciliolabrum*	TK186048	21.8x	SRR17555656	21,907	0.0043
*M. davidii*	wgs *	20.2x	ALWT00000000	18,273	0.0005
*M. occultus*	TK186204	22.9x	SRR17555655	22,220	0.0027
*M. septentrionalis*	RDS8624	14.4x	SRR17555654	22,098	0.0046
*M. thysanodes*	TK186219	24.0x	SRR17555653	21,840	0.0038
*M. velifer*	TK167846	31.9x	SRR17555652	22,070	0.0048
*M. vivesi*	NK5109	22.2x	SRR17555651	21,892	0.0039
*M. yumanensis*	TK186200	19.7x	SRR17555650	21,953	0.0041

* The wgs identifier is used throughout this study to identify the two samples from previous studies.

**Table 2 genes-13-00399-t002:** Proposed introgression pathways within *Myotis* based on 183 quartets showing significant frequency deviations from expectations under ILS.

Introgression Pathway	Pathway Number	Quartets Supporting
*M. davidii* ↔ ancestor a	I	34
*M. brandtii* ↔ ancestor c	II	55
ancestor b ↔ ancestor c	III	28
*M. brandtii* ↔ ancestor b	IV	9
*M. austroriparius* ↔ ancestor d	V	8
*M. austroriparius* ↔ *M. septentrionalis*	VI	12
*M. lucifugus* ↔ *M. septentrionalis*	VII	9
*M. thysanodes* ↔ *M. ciliolabrum*	VIII	7
*M. austroriparius* ↔ *M. lucifigus*	VIIII	13
*M. austroriparius* ↔ *M. velifer*	X	8

## Data Availability

Sequences generated for this study have been deposited in NCBI GenBank under Accession Numbers SRR17555650–SRR17555657. Custom scripts, example nexus files, a nexus file with all phylogenetically informative ves insertions from this study, and a genotype table with insertion locations and genotypes are available at https://github.com/davidaray/retrophylogenomic_tools.

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
