# Peer review of "SINE-Based Phylogenomics Reveal Extensive Introgression and Incomplete Lineage Sorting in *Myotis"

_genes, 2022, doi:10.3390/genes13030399_

Round 1

Reviewer 1 Report

The research team addresses phylogenetic issues in the Myotis genus of bats. The phylogeny of the genus remains unclear due to incomplete lineage sorting and gene flow between the species.

    In the article, authors used a number of credible methods for phylogenetic analysis that together present sufficient information to support the authors’ conclusions. Importantly, on the example of SINEs, the authors showed how the use of mobile genetic elements can facilitate phylogenetic analysis of closely-related species.

    Although, the main fundamental problem – unambiguously resolved phylogeny of the Myotis genus - remains (as it is pinpointed by the authors’ themselves), the obtained data together with the analysis of the available literature sources is trustworthy and indicates the presence of introgression between the Myotis species that can explain why it is so challenging to delineate the Myotis species.

A minor comment:

Please check the reference [1] – there is no journal and date of publishing indicated.

Author Response

Please check the reference [1] – there is no journal and date of publishing indicated.

We have added the requested information.

Reviewer 2 Report

Korstian et al. use SINE insertions and other genomic information to reconstruct phylogeny and introgression patterns in the bat clade Myotis. Generally, the paper is well written, clearly stated, and the conclusions are consistent with the analyses and results described in the paper. It is a solid dataset and analysis, which add to the growing evidence for the importance of both gene flow in mammalian evolution and the productive usage of transposon insertions as character data gleaned from short read genomes. Comments below are things that the authors might consider when revising their manuscript.

line 27. In the introductory paragraph, there are a few things to consider. Recently, authors have distinguished homoplasy (parallel/convergent evolution of a character or reversal in a character) from hemiplasy (conflict due to ILS) and from conflict due to gene flow/introgression/hybridization (I think in a recent paper, someone has given this type of conflict its own name?). In the first paragraph, the authors might use this recently developed framework (and throughout the paper), or could at least note that convergence and reversal also contribute to conflicts in phylogenetic analysis at rapid radiations. In fact, this is one major rationale for using low homoplasy transposon insertions as the authors note later in the Intro. Also, the authors might mention here that large population size increases ILS as well, assuming neutrality, in addition to short internodes.

line 42. The statement "studies have shown that concatenation methods are outperformed by methods using many gene trees to infer the overall species tree" needs citations (and more importantly, citations that are accurate and correct, which I think do not really exist...). Many simulations do not support this view (see Gatesy and Springer, 2014 in MPE for an early review), and many later simulation papers that are highly cited are fatally flawed from the start by favoring coalescent methods over concatenation in an unfair manner (see Springer and Gatesy, 2016; MPE). My impression is that it is generally the opposite upon rigorous examination, especially at deep divergences where gene tree reconstruction error is high, rather than shallow divergences with say 5 taxa or so. Limited simulations with few taxa should not be interpreted as general predictors of a phylogenomic method's utility when the simulations do not cover areas of the overall 'tree-space' where damaging biases of particular methods can occur - in particular, large gene trees with many taxa can have many incongruent nodes that bias results, while small gene trees with a few taxa cannot be outliers. For example, Bayzid and Warnow (2013; Bioinformatics) simulated fairly deeply diverging species trees with 11-17 taxa for 5-100 genes and found that MP-EST was uniformly bested by ML concatenation. Subsequently, Mirarab and Warnow (2015; Syst Biol) simulated still larger datasets. ML concatenation generally outperformed MP-EST, NJst, and often ASTRAL for large species trees (100-500 taxa) with relatively deep divergences (2M-10M generations) when 1000 loci were sampled (their figure 2), presumably due to high reconstruction error in large gene trees (their figure 5). These results contrasted with earlier simulation work based on small species trees with shallow divergences, wherein multiple hits at nucleotide sites are rare and gene tree reconstruction error is low (e.g., Kubatko et al., 2009; Liu and Edwards, 2009; Liu et al., 2009b, 2010; Liu and Yu, 2011; Hovmöller et al., 2013). So, I don't think the general statement made here without citation is even true, but if the authors assert that it is, they would maybe need to do a fairly detailed survey of simulation results where both concatenation and coalescence were done, assess the relative success of each under different circumstances, and then cite all or most relevant studies where both methods were done correctly. Some of the most cited studies that seemed to favor coalescence over concatenation, done by Mirarab and colleagues' various papers, are circular and logically flawed (as they guard against gene tree reconstruction error which a priori favors coalescence methods over concatenation - see Springer and Gatesy, 2016). Edwards et al. (2016; MPE) have argued that any success of concatenation over coalescent methods is due to concatenation being more accurate for small samples of data. But of course this cannot explain results like those summarized above, where concatenation wins even when 1000 loci are sampled (Mirarab and Warnow (2015); results like this were willfully ignored or simply missed due to poor scholarship. So, could cite Edwards et al. (2016; MPE) for noting that concatenation is bested generally in simulations (or other shallow papers like this...), but this would of course just be bad scholarship to cite prior bad scholarship and boosting a falsehood, when the goal of science is the opposite. An even cursory review of the published simulation papers that actually compared concatenation to coalescent phylogenomic methods does not show any general superiority of either method, and it is quite clear that which method does well is situational (e.g., how many taxa sampled, how deep divergences are, how many genes simulated, how fast the rate of substitution is, how concatenation was run, which coalescent method was used, how rapid radiations are in the simulated trees, at what depth the radiation happened), and really, not many situations have been done yet that compare the two approaches across a broad range of realistic conditions.

line 53. Maybe change "because they rarely experience excisions" to "because they rarely experience precise excisions". Transposons are probably commonly deleted (excised?) often from genomes in large deletions, but precise excision at the 5' and 3' ends of a single transposon seem to be rare?

Figure 1. Instead of red lines connecting taxa, a simpler and more efficient way to express conflicts of the mtDNA tree with the nuclear tree would be to simply show the two trees and put red dots at nodes that conflict in the two trees. As is, it is not really clear what the red lines indicate. For example, the clade of M. septentrionalis + M. thysanodes in the nuclear tree conflicts with the nuclear tree, but only M. septentrionalis is connected by red lines across the two trees. Presumably, the red lines indicate conflict of some sort regarding this species, but the red lines do not actually indicate which clades conflict, which is the main point of the figure?

line 78. This paragraph includes a summary of clades of Myotis with an Old World clade and a New World clade, but it seems a bit weird that the 'New World' clade includes two Old World species. Should the New World clade instead be described as a 'mostly New World' clade instead so as to avoid confusion to the reader or is it traditional to call these clades New World and Old World in the old and newer literature on Myotis systematics?

line 84. Could the unusual position of the enigmatic clade be due simply to a bat flying to a new geographic locale and establishing itself there rather than ILS or hybridization or simply be due to more mundane phylogenetic reconstruction error?

line 92. Change "in maternity colony" to "in a maternity colony"?

line 151. Should clarify what mean by "polymorphic Ves insertions". 'Polymorphic' is often used to describe variation within a species (?), but here it seems to be used to variation at a locus across species? Also on line 152, heterozygosity is mentioned. Were loci that had ambiguity in character codes due to heterozygosity included in analyses and how was the heterozygosity coded (e.g., as '?' or was one allele chosen at random for each included species when heterozygosity was apparent based on the 15X coverage)? Or, were SINE loci withe heterozygosity in any included species deleted and not included in phylogenetic analyses?

line 160. Where say, " We selected the Dollo parsimony evolutionary model because it best matches TE insertion dynamics", could also note here that recent simulations by Molloy et al. (2022 - online early at Syst Biol) under neutral conditions and short internodes show that Dollo parsimony performs better than other parsimony methods, but not as well as various coalescent methods, such as ASTRAL_BP (line 178). It is likely that Dollo is not statistically consistent for certain anomaly zone situations bases on this work.

line 168. It generally is better justified to map the bootstrap replicates onto the optimal tree(s) for the original transposon insertion matrix (i.e., instead of making a majority rule consensus of bootstrap pseudoreplicates). The bootstrap consensus can sometimes disagree with the optimal parsimony tree for a dataset. This leads to a situation where clades 'supported' by the bootstrap consensus are not even really supported by the original data (that are not resampled with replacement). Perhaps the trees here are identical, but if not, should consider mapping support scores on the optimal Dollo tree or a strict consensus of the optimal trees.

line 198. "ASTRAL_MP" should be "ASTRAL_BP" here and in other places in the text?

line 242. It is nice that the internal check here worked well.

line 261. In this paragraph, the authors discuss which trees better reflect the data, which show many conflicts, but a problem here is that the methods used do not assume that any gene flow among lineages occurs. There is no easy way around this until methods that take ILS into account credibly but also incorporate gene flow (using transposons) are created? I would sort of expect all methods to fail badle given so much evidence of gene flow, so this Myotis phylogeny is a very challenging problem, and I would not bet my life on any of the trees presented... As noted above, some simulations show that ASTRAL_BP is more accurate than Dollo parsimony when no gene flow has occurred and evolution is neutral, but when these conditions are not met, all bets are off. Also see the interpretation given on line 275. When the assumptions of the method do not hold (i.e., lots of gene flow), methods will fail, unless methods are developed that simultaneously account for ILS and gene flow, such as some approaches in PhyloNet such as MPL (rooted triplets method) or equivalent quartet based methods that estimate networks (and not just trees)? For the current empirical case, the 'correct' species tree, including introgression patterns, is not known, so it is a challenge to make solid interpretations given the conflicts among methods that are observed. Even if trees based on all methods (that assume on introgression) gave the same topology, I would not be so confident that this is the right tree, given how much introgression there seems to be in this bat case.

Fig. 2. It would seem that the SINES should have less gene tree reconstruction error relative to UCEs but at many nodes, there is less conflict in terms of quartet support in the UCE tree. Does this make sense? I think it would be very informative to add to the internal nodes of the three trees in this figure (or in an online figure) how many clean (homoplasy-free) transposons support each node. This would be of interest especially given the strong conflicts between some of the trees, including the basal placement of M. thysanodes in the ASTRAL_BP tree. How many transposon cleanly support this basal placement of M. thysanodes versus the more apical placement in the other species trees?

line 275. The following sentence read a little funny to me; perhaps clarify somehow; perhaps flesh this out to two or three sentences as the mix of high and low support in the same sentence does not make clear which clades are supported well or not. "Interestingly, the ASTRAL-BP tree (Fig. 2B) has the weakest support for the three North American clades with high posterior probability scores, with all branches receiving posterior probability scores of one despite having significant quartet support for the alternate topologies at many of the branches."

Fig. 3. The NJ tree here agrees with the ASTRAL_BP tree in placing M. thysanodes relatively basal? Is this worth noting in the text? This is a very different placement relative to the pattern for ASTRAL analysis of UCEs and for Dollo parsimony of SINEs.

line 295. Although there are many fewer incongruent SINE loci for the Dollo tree, it is possible that each of the incongruent loci implies many more extra 'steps' on the tree, on average, relative to ASTRAL_BP of SINEs and ASTRAL of UCEs. Given that Dollo is a parsimony method, I would expect it to have a cleaner fit to the tree with less steps, but this does not mean it is necessarily a good tree (e.g., see simulations in Molloy et al., in press, where 'shorter' parsimony trees are less accurate than ASTRAL_BP trees that have more steps).

Fig. 4. This is a visually nice and very informative figure. But, given gene flow, some of the underlying assumptions of the figure do not hold regarding depth of the 'synapomorphic' transposon inserts? This might be noted in the legend?

line 311. Should "114.78 bp" instead be "1147.8 bp" given the min of 655 bp and the max of 1791 bp?

line 325. Is the signal of gene flow between Old and New World bats seen in SINEs also replicated in the UCE gene trees in any significant way?   The branch lengths in Fig. 2C look very long for the branches that separate Old from New World species. How many gene trees do not show a clean separation between Old and New World species? Are the gene trees enriched for the three gene flow pathways seen in the SINE data that link some New World and Old World species together by derived transposon insertions (Fig. 5)?

line 344. How is it known that only the UCE tree has reliable branch length information? The NJ tree (Fig. 3), the network (Fig. 4A), and the ASTRAL_BP tree (Fig. 2B) all seem to put M. thysanodes basal, but this species is in a clade with M. sept... and M. cilio... and the clade of three species has a very long stem branch according to the UCE tree (Fig. 2C). For the three main 'clades' proposed, only M. austro... plus M. vel... plus M. yum... has a long stem branch according to the network tree that shows extensive conflicts for the other two main 'clades' proposed (Fig. 4A)? I think given the extent of hybridization, none of the methods used here (which all assume no gene flow) would be expected to yield reliable branch lengths, topology, or support scores, and the network in Fig. 4A sort of shows the high degree of conflict to the point where two of the three primary clades are obscured.

line 427. Some problem with the author line for this paper?
